# Subspace Attack: Exploiting Promising Subspaces for Query-Efficient Black-box Attacks

**Ziang Yan[1,3]\*    Yiwen Guo[2,3]\*    Changshui Zhang[1]**
[1]Institute for Artificial Intelligence, Tsinghua University (THUAI),
State Key Lab of Intelligent Technologies and Systems,
Beijing National Research Center for Information Science and Technology (BNRist),
Department of Automation,Tsinghua University, Beijing, China
[2] Bytedance AI Lab    [3] Intel Labs China
yza18@mails.tsinghua.edu.cn  guoyiwen.ai@bytedance.com  zcs@mail.tsinghua.edu.cn

## Abstract

Unlike the white-box counterparts that are widely studied and readily accessible, adversarial examples in black-box settings are generally more Herculean on account of the difficulty of estimating gradients. Many methods achieve the task by issuing numerous queries to target classification systems, which makes the whole procedure costly and suspicious to the systems. In this paper, we aim at reducing the query complexity of black-box attacks in this category. We propose to exploit gradients of a few reference models which arguably span some promising search subspaces. Experimental results show that, in comparison with the state-of-the-arts, our method can gain up to $2\times$ and $4\times$ reductions in the requisite mean and medium numbers of queries with much lower failure rates even if the reference models are trained on a small and inadequate dataset disjoint to the one for training the victim model. Code and models for reproducing our results are available at https://github.com/ZiangYan/subspace-attack.pytorch.

## 1   Introduction

Deep neural networks (DNNs) have been demonstrated to be vulnerable to *adversarial examples* [37] that are typically formed by perturbing benign examples with an intention to cause misclassifications. According to the amount of information that is exposed and possible to be leveraged, an intelligent adversary shall adopt different categories of attacks. Getting access to critical information (e.g., the architecture and learned parameters) about a target DNN, the adversaries generally prefer *white-box attacks* [37, 7, 24, 2, 23]. After a few rounds of forward and backward passes, such attacks are capable of generating images that are perceptually indistinguishable to the benign ones but would successfully trick the target DNN into making incorrect classifications. Whereas, so long as little information is exposed, the adversaries will have to adopt *black-box attacks* [28, 22, 3, 25, 13, 26, 38, 14, 8] instead.

In general, black-box attacks require no more information than the confidence score from a target and thus the threat model is more realistic in practice. Over the past few years, remarkable progress has been made in this regard. While initial efforts reveal the transferability of adversarial examples and devote to learning substitute models [28, 22], recent methods focus more on gradient estimation accomplished via zeroth-order optimizations [3, 25, 13, 26, 38, 14]. By issuing classification queries to the target (a.k.a., victim model), these methods learn to approach its actual gradient w.r.t. any input, so as to perform adversarial attacks just like in the white-box setting. Despite many practical merits, high query complexity is virtually inevitable for computing sensible estimations of input-gradients in some methods, making their procedures costly and probably suspicious to the classification system.

Following this line of research, we aim at reducing the query complexity of the black-box attacks. We discover in this paper that, it is possible that the gradient estimations and zeroth-order optimizations can be performed in subspaces with much lower dimensions than one may suspect, and a principled way of spanning such subspaces is considered by utilizing "prior gradients" of a few reference models as heuristic search directions. Our method, for the first time, bridges the gap between transfer-based attacks and the query-based ones. Powered by the developed mechanism, we are capable of trading the attack failure rate in favor of the query efficiency reasonably well. Experimental results show that our method can gain significant reductions in the requisite numbers of queries with much lower failure rates, in comparison with previous state-of-the-arts. We show that it is possible to obtain the reference models with a small training set disjoint to the one for training CIFAR-10/ImageNet targets.

## 2   Related Work

One common and crucial ingredient utilized in most white-box attacks is the model gradient w.r.t the input. In practical scenarios, however, the adversaries may not be able to acquire detailed architecture or learned parameters of a model, preventing them from adopting gradient-based algorithms directly. One initial way to overcome this challenge is to exploit *transferability* [37]. Ever since the adversarial phenomenon was discovered [37, 7], it has been presented that adversarial examples crafted on one DNN model can probably fool another, even if they have different architectures. Taking advantage of the transferability, Papernot et al. [27, 28] propose to construct a dataset which is labeled by querying the victim model, and train a substitute model as surrogate to mount black-box attacks. Thereafter, Liu et al. [22] study such transfer-based attacks over large networks on ImageNet [32], and propose to attack an ensemble of models for improved performance. Despite the simplicity, attacks function solely on the transferability suffer from high failure rates.

An alternative way of mounting black-box attacks is to perform gradient estimation. Suppose that the prediction probabilities (i.e., the confidence scores) of the victim model is available, methods in this category resort to zeroth-order optimizations. For example, Chen et al. [3] propose to accomplish this task using pixel-by-pixel finite differences, while Ilyas et al. [13] suggest to apply a variant of natural evolution strategies (NES) [33]. With the input-gradients appropriately estimated, they proceed as if in a white-box setting. In practice, the two are combined with the C&W white-box attack [2] and PGD [23], respectively. Though effective, owing to the high dimensionality of natural images, these initial efforts based on accurate gradient estimation generally require (tens of) thousands of queries to succeed on the victim model, which is very costly in both money and time. Towards reducing the query complexity, Tu et al. [38] and Ilyas et al. [14] further introduce an auto-encoding and a bandit mechanisms respectively that incorporate spatial and temporal priors. Similarly, Bhagoji et al. [26] show the effectiveness of random grouping and principal components analysis in achieving the goal.

In extreme scenarios where only final decisions of the victim model are exposed, adversarial attacks can still be performed [1, 4]. Such black-box attacks are in general discrepant from the score-based attacks, and we restrict our attention to the latter in this paper. As have been briefly reviewed, methods in this threat model can be divided into two categories, i.e., the *transfer-based* attacks (which are also known as the oracle-based attacks) and *query-based* attacks. Our method, probably for the first time, bridges the gap between them and therefore inherits the advantages from both sides. It differs from existing transfer-based attacks in a sense that it takes gradients of reference models as heuristic search directions for finite difference gradient estimation, and benefit from the heuristics, it is far more (query-)efficient than the latest query-based attacks.

## 3   Motivations

Let us consider attacks on an image classification system. Formally, the black-box attacks of our interest attempt to perturb an input $\mathbf{x} \in \mathbb{R}^n$ and trick a victim model $f : \mathbb{R}^n \to \mathbb{R}^k$ to give an incorrect prediction $\arg\max_i f(\mathbf{x})_i \neq y$ about its label $y$. While, on account of the high dimensionality of input images, it is difficult to estimate gradient and perform black-box attacks within a few queries, we echo a recent claim that the limitation can be reasonably ameliorated by exploiting prior knowledge properly [14]. In this section, we will shed light on the motivations of our method.

**Attack in Linear Subspaces?**   Natural images are high-dimensional and spatially over-redundant, which means not all the pixels (or combinations of pixels) are predictive of the image-level labels. A

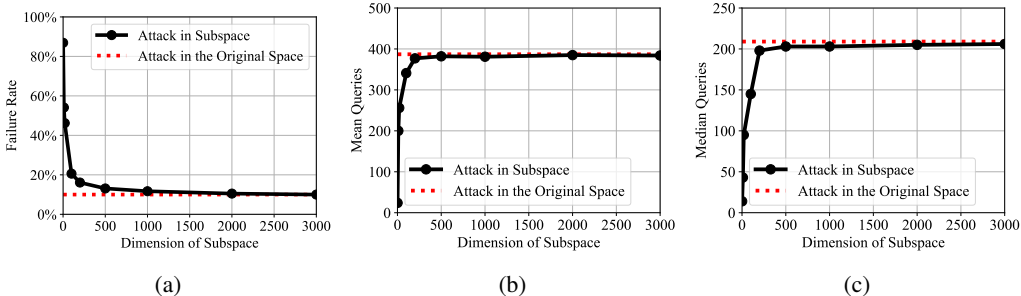

Figure 1: Black-box attack in low-dimensional random subspaces.

classification model offers its predictions typically through mining discriminative components and suppressing irrelevant variations from raw images [19]. One reasonable hypothesis worth exploring in this spirit is that, it is probably less effective to perturb an image on some specific pixels (or along certain directions) when attacking a black-box model. From a geometric point of view, that said, the problem probably has a lower intrinsic dimension than $n$, just like many other ones [20].

To verify this, we try estimating gradients and mounting attacks on low-dimensional subspaces for images, which is bootstrapped by generating $m < n$ random basis vectors $\mathbf{u}_0, \ldots \mathbf{u}_{m-1}$ sequentially on condition of each being orthogonal to the prior ones. We utilize the bandit optimization advocated in a recent paper [14] for gradient estimation, and adopt the same iterative attack (i.e., PGD) as in it. Recall that the bandit mechanism updates its estimation $\mathbf{g}_t$ at each step by a scaled search direction:

$$\mathbf{\Delta}_t = \frac{l(\mathbf{g}_t + \delta \mathbf{u}_t') - l(\mathbf{g}_t - \delta \mathbf{u}_t')}{\delta} \mathbf{u}_t', \tag{1}$$

in which $\mathbf{u}_t'$ is the search direction sampled from a Gaussian distribution, $\delta > 0$ is a step size that regulates the directional estimation, and $l(\cdot)$ calculates the inner product between its normalized input and the precise model gradient. The mechanism queries a victim model twice at each step of the optimization procedure for calculating $\mathbf{\Delta}_t$, after which a PGD step based on the current estimation is applied. Interested readers can check the insightful paper [14] for more details.

In this experiment, once the basis $\{\mathbf{u}_0, \ldots \mathbf{u}_{m-1}\}$ is established for a given image, they are fixed over the whole optimization procedure that occurs on the $m$-dimensional subspace instead of the original $n$-dimensional one. More specifically, the search direction $\mathbf{u}_t'$ is yielded by combining the generated basis vectors with Gaussian coefficients, i.e., $\mathbf{u}_t' = \sum_i \alpha_i \mathbf{u}_i$ and $\alpha_i \sim \mathcal{N}(0, 1)$. We are interested in how the value of $m$ affects the failure rate and the requisite number of queries of successful attacks. By sampling 1,000 images from the CIFAR-10 test set, we craft untargeted adversarial examples for a black-box wide residual network (WRN) [41] with an upper limit of 2,000 queries for efficiency reasons. As depicted in Figure 1, after $m > 500$, all three concerned metrics (i.e., failure rate, mean and median query counts) barely change. Moreover, at $m = 2000$, the failure rate already approaches $\sim 10\%$, which is comparable to the result gained when the same optimization is applied in the original image space which has $n = 3072$ dimensions. See the red dotted line in Figure 1 for this baseline. Similar phenomenon can be observed on other models using other attacks as well, which evidences that the problem may indeed have a lower dimension than one may suspect and it complements the study of the intrinsic dimensionality of training landscape of DNNs in a prior work [20].

**Prior Gradients as Basis Vectors?**     Since the requisite number of queries at $m = 2000$ is already high in Figure 1, we know that the random basis vectors boost the state-of-the-art only to some limited extent. Yet, it inspires us to explore more principled subspace bases for query-efficient attacks. To achieve this goal, we start from revisiting and analyzing the transfer-based attacks. We know from prior works that even adversarial examples crafted using some single-step attacks like the fast gradient (sign) [18] can transfer [28, 22], hence one can hypothesize that the gradients of some "substitute" models are more helpful in spanning the search subspaces with reduced dimensionalities. A simple yet plausible way of getting these gradients involved is to use them directly as basis vectors. Note that unlike the transfer-based attacks in which these models totally substitute for the victim when crafting adversarial examples, our study merely considers their gradients as priors. We refer to such models and gradients as *reference models* and *prior gradients* respectively throughout this paper for clarity.

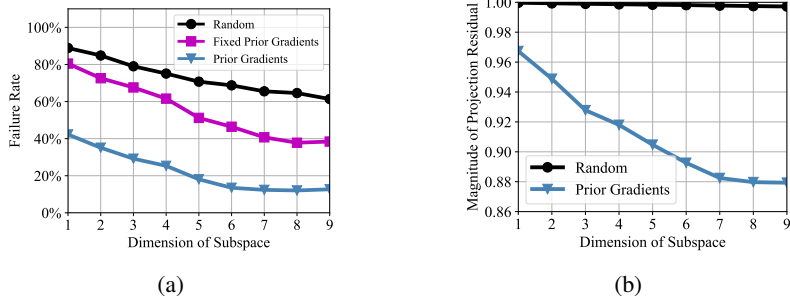

(a)                                                        (b)

Figure 2: Comparison of (a) the failure rates when attacking WRN, and (b) mean squared residuals of projecting the precise gradient onto subspaces spanned by random directions or prior gradients. We collect nine models as candidates to obtain the prior gradients: AlexNet [17], VGG-11/13/16/19 [34], and ResNet-20/32/44/56 [10]. We add prior gradients corresponding to models from deep to shallow one by one to the basis set.

The simplest solution to utilize such prior gradients might be set these basis vectors to be *fixed* over the entire optimization procedure, i.e., only the input-gradient of reference models with respect to the clean image **x** are utilized. We further let these basis vectors be adaptive when applying an iterative attack (e.g., the basic iterative method [18] and PGD [23]), simply by recalculating the prior gradients (w.r.t the current inputs which may be candidate adversarial examples) at each step. Different zeroth-order optimization algorithms can be readily involved in the established subspaces. For simplicity, we will stick with the described bandit optimization in the sequel of this paper and we leave the exploration on other algorithms like the coordinate-wise finite differences [3] and NES [13] to future works.

An experiment is similarly conducted to compare attacks in the gradient-spanned subspaces[1] and the random ones, in which the WRN is still regarded as the victim model. We compare mounting black-box attacks on different subspaces spanned by the adaptive and fixed prior gradients, as well as randomly generated vectors as described before. Figure 2 summarizes our main results. As in Figure 1(a), we illustrate the attack failure rates in Figure 2(a). Apparently, the adaptive prior gradients are much more promising than its fixed and random counterparts when spanning search subspaces. We would use the adaptive version of prior gradients in the rest of this paper. For more insights, we project normalized WRN gradients (calculated on clean images) onto the two sorts of subspaces and further compare the mean squared residuals of projection under different circumstances in Figure 2(b). It can be seen that the gradient-spanned subspaces indeed align better with the precise WRN gradients, and over misalignments between the search subspaces and precise model gradients lead to high failure rates.

## 4  Our Subspace Attack

As introduced in the previous section, we reckon that it is promising to apply the gradient of some reference models to span the search subspace for mounting black-box attacks. However, there remain some challenges in doing so. First, it should be computationally and memory intensive to load all the reference models and calculate their input-gradients as basis vectors. Second, it is likely that an "universal" adversarial example for a victim model is still far away from such subspaces, which means mounting attacks solely on them may lead to high failure rate as encountered in the transfer-based attacks. We will discuss the issues and present our solutions in this section. We codename our method subspace attack and summarize it in Algorithm 1, in which the involved hyper-parameters will be carefully explained in Section 5.

### 4.1  Coordinate Descent for Efficiency

If one of the prior gradients happens to be well-aligned with the gradient of the victim model, then "an adaptive" one-dimensional subspace suffices to mount the attack. Nevertheless, we found that it is normally not the case, and increasing the number of reference models and prior gradients facilitates

**Algorithm 1** Subspace Attack Based on Bandit Optimization [14]

---

1: **Input:** a benign example $\mathbf{x} \in \mathbb{R}^n$, its label $y$, a set of $m$ reference models $\{f_0, \ldots, f_{m-1}\}$, a chosen attack objective function $\mathcal{L}(\cdot, \cdot)$, and the victim model from which the output of $f$ can be inferred.
2: **Output:** an adversarial example $\mathbf{x}_{\mathrm{adv}}$ fulfills $\|\mathbf{x}_{\mathrm{adv}} - \mathbf{x}\|_\infty \leq \epsilon$.
3: Initialize the adversarial example to be crafted $\mathbf{x}_{\mathrm{adv}} \leftarrow \mathbf{x}$.
4: Initialize the gradient to be estimated $\mathbf{g} \leftarrow \mathbf{0}$.
5: Initialize the drop-out/layer ratio $p$.
6: **while** not successful **do**
7: $\quad$ Choose a reference model whose index is $i$ uniformly at random
8: $\quad$ Calculate a prior gradient with drop-out/layer ratio $p$ as $\mathbf{u} \leftarrow \frac{\partial \mathcal{L}(f_i(\mathbf{x}_{\mathrm{adv}};p),y)}{\partial \mathbf{x}_{\mathrm{adv}}}$
9: $\quad \mathbf{g}_+ \leftarrow \mathbf{g} + \tau \mathbf{u}, \quad \mathbf{g}_- \leftarrow \mathbf{g} - \tau \mathbf{u}$
10: $\quad \mathbf{g}'_+ \leftarrow \mathbf{g}_+/\|\mathbf{g}_+\|_2, \quad \mathbf{g}'_- \leftarrow \mathbf{g}_-/\|\mathbf{g}_-\|_2$
11: $\quad \boldsymbol{\Delta}_t \leftarrow \frac{\mathcal{L}(f(\mathbf{x}_{\mathrm{adv}}+\delta \mathbf{g}'_+),y)-\mathcal{L}(f(\mathbf{x}_{\mathrm{adv}}+\delta \mathbf{g}'_-),y)}{\tau \delta} \mathbf{u}$
12: $\quad \mathbf{g} \leftarrow \mathbf{g} + \eta_{\mathbf{g}} \boldsymbol{\Delta}_t$
13: $\quad \mathbf{x}_{\mathrm{adv}} \leftarrow \mathbf{x}_{\mathrm{adv}} + \eta \cdot \mathrm{sign}(\mathbf{g})$
14: $\quad \mathbf{x}_{\mathrm{adv}} \leftarrow \mathrm{Clip}(\mathbf{x}_{\mathrm{adv}}, \mathbf{x} - \epsilon, \mathbf{x} + \epsilon)$
15: $\quad \mathbf{x}_{\mathrm{adv}} \leftarrow \mathrm{Clip}(\mathbf{x}_{\mathrm{adv}}, 0, 1)$
16: $\quad$ Update the drop-out/layer ratio $p$ following our policy
17: **end while**
18: **return** $\mathbf{x}_{\mathrm{adv}}$

---

the attack, which can be partially explained by the fact that they are nearly orthogonal to each other in high-dimensional spaces [22]. Definitely, it is computationally and memory intensive to calculate the input-gradients of a collection of reference models at each step of the optimization.

Given a set of basis vectors, off-the-shelf optimization procedures for black-box attacks either estimate the optimal coefficients for all vectors before update [3] or give one optimal scaling factor overall [14]. For any of them, the whole procedure is somewhat analogous to a gradient descent whose update directions do not necessarily align with single basis vectors. It is thus natural to make an effort based on coordinate descent [39], which operates along coordinate directions (i.e., basis vectors) to seek the optimum of an objective, for better efficiency. In general, the algorithm selects a single coordinate direction or a block of coordinate directions to proceed iteratively. That said, we may only need to calculate one or several prior gradients at each step before update and the complexity of our method is significantly reduced. Experimental results in Section 5 show that one single prior gradient suffices.

## 4.2 Drop-out/layer for Exploration

As suggested in Figure 2(b), one way of guaranteeing a low failure rate in our method is to collect adequate reference models. However, it is usually troublesome in practice, if not infeasible. Suppose that we have collected a few reference models which might not be adequate, and we aim to reduce the failure rate whatsoever. Remind that the main reason of high failure rates is the imperfect alignment between our search subspaces and the precise gradients (cf., Figure 2(b)), however, it seems unclear how to explore other possible search directions without training more reference models. One may simply try adding some random vectors to the basis set for better alignment and higher subspace-dimensions, although they bare the ineffectiveness as discussed in Section 3 and we also found in experiments that this strategy does not help much.

Our solution to resolve this issue is inspired by the dropout [35] and "droplayer" (a.k.a., stochastic depth) [12] techniques. Drop-out/layer, originally serve as regularization techniques, randomly drop a subset of hidden units or residual blocks (if exist) from DNNs during training. Their successes indicate that a portion of the features can provide reasonable predictions and thus meaningful input-gradients, which implies the possibility of using drop-out/layer invoked gradients to enrich our search priors [2]. By temporarily removing hidden units or residual blocks, we can acquire a spectrum of prior gradients from each reference model. In experiments, we append dropout to all convolutional/fully-connect layer (except the final one), and we further drop residual blocks out in ResNet reference models.

# 5   Experiments

In this section, we will testify the effectiveness of our subspace attack by comparing it with the state-of-the-arts in terms of the failure rate and the number of queries (of successful attacks). We consider both untargeted and targeted $\ell_\infty$ attacks on CIFAR-10 [16] and ImageNet [32]. All our experiments are conducted on a GTX 1080 Ti GPU with PyTorch [29]. Our main results for untargeted attacks are summarized in Table 1, and the results for targeted attacks are reported in the supplementary material.

Table 1: Performance of different black-box attacks with $\ell_\infty$ constraint under untargeted setting. The maximum perturbation is $\epsilon = 8/255$ for CIFAR-10, and $\epsilon = 0.05$ for ImageNet. A recent paper [26] also reports its result on WRN similarly, which achieves a failure rate of 1.0% with 7680 queries. PyramidNet* in the table indicates PyramidNet+ShakeDrop+AutoAugment [5].

| Dataset | Victim Model | Method | Ref. Models | Mean Queries | Median Queries | Failure Rate |
|---------|-------------|--------|-------------|--------------|----------------|--------------|
| CIFAR-10 | WRN | NES [13] | - | 1882 | 1300 | 3.5% |
| | | Bandits-TD [14] | - | 713 | 266 | 1.2% |
| | | Ours | AlexNet+VGGNets | **392** | **60** | **0.3%** |
| | GDAS | NES [13] | - | 1032 | 800 | 0.0% |
| | | Bandits-TD [14] | - | 373 | 128 | 0.0% |
| | | Ours | AlexNet+VGGNets | **250** | **58** | **0.0%** |
| | PyramidNet* | NES [13] | - | 1571 | 1300 | 5.1% |
| | | Bandits-TD [14] | - | 1160 | 610 | 1.2% |
| | | Ours | AlexNet+VGGNets | **555** | **184** | **0.7%** |
| ImageNet | Inception-v3 | NES [13] | - | 1427 | 800 | 19.3% |
| | | Bandits-TD [14] | - | 887 | 222 | 4.2% |
| | | Ours | Original ResNets | **462** | **96** | **1.1%** |
| | PNAS-Net | NES [13] | - | 2182 | 1300 | 38.5% |
| | | Bandits-TD [14] | - | 1437 | 552 | 12.1% |
| | | Ours | Original ResNets | **680** | **160** | **4.2%** |
| | SENet | NES [13] | - | 1759 | 900 | 17.9% |
| | | Bandits-TD [14] | - | 1055 | 300 | 6.4% |
| | | Ours | Original ResNets | **456** | **66** | **1.9%** |

## 5.1   Experimental Setup

**Evaluation Metrics and Settings.**   As in prior works [13, 26, 14], we adopt the failure rate and the number of queries to evaluate the performance of attacks using originally correctly classified images. For untargeted settings, an attack is considered successful if the model prediction is different from the ground-truth, while for the targeted settings, it is considered successful only if the victim model is tricked into predicting the target class. We observe that the number of queries changes dramatically between different images, thus we report both the mean and median number of queries of successful attacks to gain a clearer understanding of the query complexity.

Following prior works, we scale the input images to $[0, 1]$, and set the maximum $\ell_\infty$ perturbation to $\epsilon = 8/255$ for CIFAR-10 and $\epsilon = 0.05$ for ImageNet. We limit to query victim models for at most 10,000 times in the untargeted experiments and 50,000 times in the targeted experiments, as the latter task is more difficult and requires more queries. In all experiments, we invoke PGD [23] to maximize the hinge logit-diff adversarial loss from Carlini and Wagner [2]. The PGD step size is set to $1/255$ for CIFAR-10 and 0.01 for ImageNet. At the end of each iteration, we clip the candidate adversarial examples back to $[0, 1]$ to make sure they are still valid images. We initialize the drop-out/layer ratio as 0.05 and increase it by 0.01 at the end of each iteration until it reaches 0.5 throughout our experiments. Other hyper-parameters like the OCO learning rate $\eta_{\mathbf{g}}$ and the finite-difference step sizes (i.e., $\delta, \tau$) are set following the paper [14]. We mostly compare our method with NES [13] and Bandits-TD [14], and their official implementations are directly used. We apply all the attacks on the same set of clean images and victim models for fair comparison. For Bandits-TD on ImageNet, we craft adversarial examples on a resolution of $50 \times 50$ and upscale them according to specific requests from the victim models (i.e., $299 \times 299$ for Inception-v3, $331 \times 331$ for PNAS-Net, and $224 \times 224$ for SENet) before query, just as described in the paper [14]. We do not perform such rescaling on CIFAR-10 since no performance gain is observed.

**Victim and Reference Models.** On CIFAR-10, we consider three victim models: (a) a WRN [41] with 28 layers and 10 times width expansion [3], which yields 4.03% error rate on the test set; (b) a model obtained via neural architecture search named GDAS [6] [4], which has a significantly different architecture than our AlexNet and VGGNet reference models and shows 2.81% test error rate; (c) a 272-layer PyramidNet+Shakedrop model [9, 40] trained using AutoAugment [5] with only 1.56% test error rate, [5] which is the published state-of-the-art on CIFAR-10 to the best of our knowledge. As for reference models, we simply adopt the AlexNet and VGG-11/13/16/19 architectures with batch normalizations [15]. To evaluate in a more data-independent scenario, we choose an auxiliary dataset (containing only 2,000 images) called CIFAR-10.1 [30] to train the reference models from scratch.

We also consider three victim models on ImageNet: (a) an Inception-v3 [36] which is commonly chosen [13, 14, 4, 38] with 22.7% top-1 error rate on the official validation set; (b) a PNAS-Net-5-Large model [21] whose architecture is obtained through neural architecture search, with a top-1 error rate of 17.26%; (c) an SENet-154 model [11] with a top-1 error rate of 18.68% [6]. We adopt ResNet-18/34/50 as reference architectures, and we gather 30,000+45,000 images from an auxiliary dataset [31] and the ImageNet validation set to train them from scratch. The clean images for attacks are sampled from the remaining 5,000 ImageNet official validation images and hence being unseen to both the victim and reference models.

## 5.2 Comparison with The State-of-the-arts

In this section we compare the performance of our subspace attack with previous state-of-the-art methods on CIFAR-10 and ImageNet under untargeted settings.

On CIFAR-10, we randomly select 1,000 images from its official test set, and mount all attacks on these images. Table 1 summarizes our main results, in which the fifth to seventh columns compare the mean query counts, median query counts and failure rates. On all three victim models, our method significantly outperforms NES and Bandits-TD in both query efficiency and success rates. By using our method, we are able to reduce the mean query counts by a factor of 1.5 to 2.1 times and the median query counts by 2.1 to 4.4 times comparing with Bandits-TD which incorporates both time and spatial priors [14]. The PyramidNet+ShakeDop+AutoAugment [5] model, which shows the lowest test error rate on CIFAR-10, also exhibits the best robustness under all considered black-box attacks. More interestingly, even if the victim model is GDAS, whose architecture is designed by running neural architecture search and thus being drastically different from that of the reference models, our prior gradients can still span promising subspaces for attacks. To the best of our knowledge, we are the first to attack PyramidNet+ShakeDrop+AutoAugment which is a published state-of-the-art and GDAS which has a searched architecture in the black-box setting.

For ImageNet, we also randomly sample 1,000 images from the ImageNet validation set for evaluation. Similar to the results on CIFAR-10, the results on ImageNet also evidence that our method outperforms the state-of-the-arts by large margins. Moreover, since the applied reference models are generally more "old-fashioned" and computationally efficient than the victim models that are lately invented, our method introduces little overhead to the baseline optimization algorithm.

## 5.3 Dropout Ratios and Training Scales

We are interested in how the dropout ratio would affect our attack performance. To figure it out, we set an upper limit of the common dropout ratio $p$ to 0.0, 0.2, 0.5 respectively to observe how the query complexity and the failure rate vary when attacking the WRN victim model. With the AlexNet and VGGNet reference models trained on CIFAR-10.1 [30], we see from the bottom of Table 2 that more dropout indicates lower failure rate, verifying that exploration via dropout well amends the misalignments between our subspaces and the victim model gradients.

It might also be intriguing to evaluate how the performance of our method varies with the scale of training set for yielding reference models. We attempt to evaluate it empirically by training AlexNet and VGGNets from scratch using different numbers of training images. More specifically,

Table 2: Impact of the dropout ratio and training scale on CIFAR-10. The victim model is WRN.

| Ref. Training Set | #Images | Maximum $p$ | Mean Queries | Median Queries | Failure Rate |
|---|---|---|---|---|---|
| CIFAR-10 Training | 50k | 0.0 | 59 | 12 | 1.4% |
| | | 0.2 | 77 | 14 | 0.2% |
| | | 0.5 | 111 | 14 | 0.2% |
| CIFAR-10.1 + CIFAR-10 Test (Part) | 2k+8k | 0.0 | 239 | 16 | 3.2% |
| | | 0.2 | 174 | 20 | 0.7% |
| | | 0.5 | 212 | 22 | 0.3% |
| CIFAR-10.1 | 2k | 0.0 | 519 | 48 | 9.6% |
| | | 0.2 | 380 | 62 | 0.9% |
| | | 0.5 | 392 | 60 | 0.3% |

we enlarge our training set by further using the CIFAR-10 official training and test images, *excluding the 1,000 images for mounting attacks of course*. In addition to the CIFAR-10.1 dataset as used, we try two larger sets: (a) the official CIFAR-10 training set which consists of 50,000 images; [7] (b) a set built by augmenting CIFAR-10.1 with 8,000 CIFAR-10 test images, whose overall size is 2,000+8,000=10,000. It can be seen from Table 2 that by training reference models with 8,000 more images, the query counts could be cut by over $2\times$ without dropout, and the failure rate decreases as well. We believe that the performance gain is powered by better generalization ability of the reference models. In a special scenario where the reference and the victim models share the same training set, our method requires only 59 queries on average to succeed on 98.6% of the testing images without dropout. The performance of our method with dropout is also evaluated on the basis of these reference models, and we can see that dropout is capable of reducing the failure rates significantly regardless of the reference training set. While for the query complexity, we may observe that more powerful reference models generally require less exploration governed by dropout to achieve efficient queries.

## 5.4   Choice of Reference Models and Prior Gradients

Table 3: Subspace attack using different reference models with $\ell_\infty$ constraint under untargeted setting on CIFAR-10. The maximum perturbation is $\epsilon = 8/255$, and the victim model is WRN.

| Ref. Models | Mean Queries | Median Queries | Failure Rate |
|---|---|---|---|
| VGG-19 | 400 | 78 | 0.6% |
| VGG-19/16/13 | 395 | 71 | 0.4% |
| VGG-19/16/13/11+AlexNet | 392 | 60 | 0.3% |

We investigate the impact of number and architecture of reference models for our method by evaluating our attack using different reference model sets, and report the performance in Table 3. As in previous experiments, reference models are trained on CIFAR-10.1, and the maximum dropout ratio is set to 0.5. We see that increasing the number of reference models indeed facilitates the attack in both query efficiency and success rates, just like in the exploratory experiment where dropout is absent.

We also compare using "gradient descent" and "coordinate descent" empirically. On CIFAR-10 we choose the same five reference models as previously reported, and at each iteration we compute all five prior gradients and search in the complete subspace. We combine all the prior gradients with Gaussian coefficients to provide a search direction in it. Experimental results demonstrate that with significantly increased run-time, both the query counts and failure rates barely change (mean/median queries: 389/62, failure rate: 0.3%), verifying that our coordinate-descent-flavored policy achieves a sensible trade-off between efficiency and effectiveness.

## 6   Conclusion

While impressive results have been gained, state-of-the-art black-box attacks usually require a large number of queries to trick a victim classification system, making the process costly and suspicious to

the system. In this paper, we propose the subspace attack method, which reduces the query complexity by restricting the search directions of gradient estimation in promising subspaces spanned by input-gradients of a few reference models. We suggest to adopt a coordinate-descent-flavored optimization and drop-out/layer to address some potential issues in our method and trade off the query complexity and failure rate. Extensive experimental results on CIFAR-10 and ImageNet evidence that our method outperforms the state-of-the-arts by large margins, even if the reference models are trained on a small and inadequate dataset disjoint to the one for training the victim models. We also evaluate the effectiveness of our method on some winning models (e.g., PyramidNet+ShakeDrop+AutoAugment [5] and SENet [11]) on these datasets and models whose architectures are designed by running neural architecture search (e.g., GDAS [6] and PNAS [21]).

### Acknowledgments

This work is funded by NSFC (Grant No. 61876095) and Beijing Academy of Artificial Intelligence (BAAI).

## Footnotes

*The first two authors contributed equally to the work. Work was done when YG was with Intel Labs China.

[1]Granted, the prior gradients are almost surely linearly independent and thus can be regarded as basis vectors.

[2]We examine the generated input-gradients in this manner and found that most of them are still independent.

[3]Pre-trained model: https://github.com/bearpaw/pytorch-classification

[4]Pre-trained model: https://github.com/D-X-Y/GDAS

[5]Unlike the other two models that are available online, this one is trained using scripts from: https://github.com/tensorflow/models/tree/master/research/autoaugment

[6]Pre-trained models: https://github.com/Cadene/pretrained-models.pytorch

[7]In this special setting the reference models and the victim model share the same training data.

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
