[Supplementary Material]

# Subspace Attack: Exploiting Promising Subspaces for Query-Efficient Black-box Attacks Supplementary Material

**Ziang Yan**[1,3*]  **Yiwen Guo**[2,3*]  **Changshui Zhang**[1]

[1]Institute for Artificial Intelligence, Tsinghua University (THUAI),
State Key Lab of Intelligent Technologies and Systems,
Beijing National Research Center for Information Science and Technology (BNRist),
Department of Automation,Tsinghua University, Beijing, China
[2] Bytedance AI Lab   [3] Intel Labs China
yza18@mails.tsinghua.edu.cn  guoyiwen.ai@bytedance.com  zcs@mail.tsinghua.edu.cn

## 1  Results for Targeted Attacks

Table 1: Performance of different black-box attacks with $\ell_\infty$ constraint under the targeted setting on CIFAR-10. The maximum perturbation is $\epsilon = 8/255$. A recent paper [6] also reports its result on WRN similarly, which achieves a failure rate of 6.0% with 7680 queries. PyramidNet* in the table indicates PyramidNet+ShakeDrop+AutoAugment [3]. Reference models are the same as in the paper.

| Victim Model | Method | Mean Queries | Median Queries | Failure Rate |
|---|---|---|---|---|
| WRN | NES [4] | 6007 | 3600 | 9.6% |
| | Bandits-TD [5] | 5003 | 1510 | 6.3% |
| | Ours | **4360** | **690** | **4.4%** |
| GDAS | NES [4] | 2062 | 1500 | 0.0% |
| | Bandits-TD [5] | 1707 | 942 | 0.0% |
| | Ours | **1452** | **604** | **0.0%** |
| PyramidNet* | NES [4] | 4468 | 2700 | 2.4% |
| | Bandits-TD [5] | 4881 | 2344 | 2.0% |
| | Ours | **3495** | **1334** | **1.6%** |

In this section we evaluate our subspace attack on the targeted setting. The target class for each image to be tested is uniformly sampled from all possible classes except the ground-truth class. Different from the untargeted setting, a targeted attack is counted as successful only if the model is tricked into predicting the target class, so in general the task is more difficult than its untargeted counterpart. We found Bandits-TD with the original OCO learning rate $\eta_{\mathbf{g}} = 100$ in [5] has even worse performance than NES on all three architectures in this task, so we perform a grid search on other 100 test set images (other than the 1,000 images to be tested) for $\eta_{\mathbf{g}}$ in $\{0.1, 1, 10, 100\}$ and find that $\eta_{\mathbf{g}} = 1$ leads to the best performance for Bandits-TD. Then for targeted attacks we set $\eta_{\mathbf{g}} = 1$ for both Bandits-TD and our subspace attack for a fair comparison.

The results on CIFAR-10 are reported in Table 1. Similar to the untargeted setting, it can be observed that our proposed method significantly reduces the query complexity and the failure rates on all victim models. Although previous transfer-based attacks often have high failure rates in targeted attacks, our results demonstrate that if and when utilized properly, reference models can indeed provide valuable information to attack the victim model, regardless of untargeted or targeted. Notice that just like in the untargeted setting, our performance gain on the two prevalent metrics for query complexity (i.e.,

mean and median numbers) are different in Table 1, which demonstrates that the required number of queries changes dramatically on different images and it encourages us to bring some other sample statistics for a more comprehensive comparison in future works.

We future testify $\ell_\infty$ targeted attack using our method on ImageNet. The PGD step size and maximum perturbation limit are kept the same as in ImageNet untargeted experiments, and our method achieves 0.8% failure rate using 7,695/3,200 mean/median queries.

## 2   Attack Models Guarded via Ensemble Adversarial Training

Table 2: Performance of different black-box attack methods when attacking ensemble adversarially trained InceptionV3 [7] with $\ell_\infty$ constraint under untargeted setting on ImageNet. The maximum perturbation is $\epsilon = 0.05$. Reference models are kept the same as in the main paper.

| Victim Model | Method | Mean Queries | Median Queries | Failure Rate |
|---|---|---|---|---|
| | NES [4] | 1445 | 800 | 30.5% |
| Adv. Inception-v3 [7] | Bandits-TD [5] | 931 | 270 | 3.2% |
| | Ours | **607** | **96** | **2.6%** |

In this section we further verify the effectiveness of our method on a victim model which is trained using ensemble adversarial training [7]. Since it has been criticized that transfer-based attacks may encounter problems in attacking victim models guarded via adversarial training, we would like to check the performance of our method accordingly. Here we directly use a pre-trained Inception-v3 model released officially [1], which is trained with Step-LL examples generated on an ensemble of four models [7]. Table 2 summarizes our results. We see on the adversarially trained victim model our subspace attack still outperforms NES and Bandits-TD in both query-efficiency and failure rate, although our reference models are naturally trained.

## 3   Results for $\ell_2$ Attacks

Table 3: Performance of different black-box attack methods under the $\ell_2$ targeted setting on CIFAR-10. The maximum $\ell_2$ perturbation is $\epsilon = 4.6$, which corresponds to per-pixel distortion 0.0015 as reported in the AutoZOOM [8] paper. Reference models are kept the same as in the main paper.

| Victim Model | Method | Mean Queries | Median Queries | Failure Rate |
|---|---|---|---|---|
| | ZOO [1] | 10,784 | - | >3.0% |
| | ZOO+AE [1] | 5,378 | - | >1.0% |
| | AutoZOOM-BiLIN [8] | 835 | - | >0.7% |
| ConvNet [1, 8] | AutoZOOM-AE [8] | 345 | - | >0.0% |
| | NES [4] | 976 | 800 | 0.0% |
| | Bandits-TD [5] | 206 | 140 | 0.0% |
| | Ours | **69** | **34** | **0.0%** |

In this section we use the same 1,000 CIFAR-10 images to test as in previous experiments, and we evaluate different black-box attacks under the $\ell_2$ targeted setting on them. To make a fair comparison with the state-of-the-arts, we use the same victim model (named "ConvNet" in Table 3) for testing NES, Bandits-TD, and our method as in ZOO [1] and AutoZOOM [8], which has seven weight layers and a test error rate of 22.03%. The model is pre-trained by Carlini and Wagner [2], and the $\ell_2$ perturbation budget for NES, Bandits-TD, and our method is set to be $\epsilon = 4.6 = 3072 \times 0.0015$. We directly cite ZOO and AutoZOOM results from the paper [8]. Note since they do not enforce a strict maximum perturbation budget, they should achieve no greater than their reported attack success rates once the $\epsilon = 4.6$ perturbation limit is set [5]. For NES and Bandits-TD results, we set $\eta_\mathbf{g} = 0.01$, $\tau = \delta = 1.0$, and $\eta = 0.5$ after grid search on some other 100 images from the official CIFAR-10 test set which are different from the 1,000 images for performance evaluation and are unseen to both

the victim model and reference models. For our method, we adopt the same hyper-parameters as for Bandits-TD. The query limit is set to 50,000 times, following our previous targeted experiments.

Results for the $\ell_2$ targeted attacks on CIFAR-10 are summarized in Table 3. We see in this setting our method again outperforms all competitive methods by large margins in both query efficiency and failure rates, validating its effectiveness.

## 4   The Threat Model

Since we are the (tied) first to take reference models into consideration when performing query-based attacks [2], we follow prior arts and mainly compare the number of mean and medium queries. Yet, we feel that it is also essential to consider the cost of training and querying reference models under the threat model, which is a bit subjective though. We advocate run-time comparisons for querying references in follow-up works, and our well-trained reference models will be available online as well.

## Footnotes

*The first two authors contributed equally to the work. Work was done when YG was with Intel Labs China.

[1]`https://github.com/tensorflow/models/tree/master/research/adv_imagenet_models`

[2]`https://github.com/carlini/nn_robust_attacks`