[Reviews · NeurIPS 2019]

Reviewer 1



- The authors proposed a heuristic black-box attack algorithm which improves the query efficiency of black-box attacks by utilizing prior gradients from reference models. The assumption is that gradients from the reference models should also transfer to the target model and thus could help better estimating the true gradient. - The paper did not comes with any theoretical justifications. It is fine for a heuristic method, yet I would expect some more direct evidence to support the key arguments here. In this case, I think the authors made an important assumption implicitly, i.e., the gradient of the reference models should also transfer to the target model just like those adversarial examples (which are shown to have some transferability between models). However, I did not find any direct evidence or supportive experiments to convince the reader that it is indeed true. For example, the authors can consider comparing the cosine similarity between reference model’s gradients and the true gradients, with the cosine similarity through random gaussians and show that the reference model indeed provides more accurate gradient search directions. - SUBSPACE ATTACK? The proposed method is called subspace attack. I guess it is because as the authors showed in Figure 1, using m (less than n) basis vectors, one could still achieve similar attack performances. It is nice to know that but i do not think it is related to the proposed method as in this case the basis vectors are fixed during the experiments while the proposed method uses reference model’s gradient which changes at different points. I think the key idea here is about using reference model’s gradients as basis vectors, not about using less than n basis vectors. From this point of view, I think Figure 1 and “subspace attack” name is a bit misleading. - Suppose the above mentioned transfer assumption is indeed true. The proposed method is working only if we had the reference models at the first place (In other words, only if we know the data distribution first). I would be nice for the authors to consider the case where we do not know the data distribution or at least comment on it. Minor comments: - Line 124, what is adaptive mean, i did not get it clearly - Section 4.1, why it is coordinate descent? I think ZOO method [3] is in the coordinate descent fashion as it only changes one dimension at each step. The proposed method seems not to be this case? Or you just want to say it uses one search direction at each step? ======================= I have read the authors' response and it clears most of my concerns and hence I increase the score

Reviewer 2



The idea of using reference models to guide query-based search is original and quite important in bridging the two lines of approaches to black-box attacks. The algorithm is explained clearly and its performance is evaluated with thorough experiments. The significance of the moving parts of the algorithm is also evaluated with carefully designed experiments. However, I have one major concern with the current draft: Whenever we propose a method in adversarial ML or any security domain in general, it is of utmost importance to first define the threat model clearly. The threat model typically includes a number of components: 1. What's the attacker's goal? Targeted attack or untargeted attack? How many points does he need to attack (important)? 2. What information/resource does the attacker have? Is he given a reference model for free? Or does the reference model come in the cost of a certain number of queries, e.g. the number of training points used to train it? 3. What can the attacker do, e.g. what's the interaction protocol between the attacker and the victim learner? e.g. White-box or Black-box. 4. What's the cost of the attacker? e.g. Number of queries. Depend on the choice of the threat model, the evaluation standard should be different. When comparing different algorithms, it is important to compare them under the same threat model, in which all attackers, though equipped with different attack methods, have access to the same resources. In this particular paper, it is not fair to compare the proposed method, which utilizes a valuable resource, the reference model, with methods that don't, e.g. NES and Bandits-TD. A fair comparison should take into consideration of the potential cost of this reference model. For example, one can consider the cost of the reference model as spending a certain amount of queries as overhead training points for the reference model. Then, depending on the total number of examples the attacker needs to attack (specified in the threat model), this number can be divided between all examples. In the CIFAR10 experiment, the attacker attacks 1000 points, with 2000 training points for the reference model, so the mean and median number of queries should increase by 2 = 2000/1000 to account for this overhead. Then, there will be a trade-off and a choice to be made. When the number of points to be attacked is small, the attacker may not want to spend a large number of queries to train a reference model, whereas when the number of target points is enormous, it is a good strategy to be considered. This is the principled way of doing researches on problems in the adversarial setting, and is highly recommended.

Reviewer 3



The paper improved a previous score-based blackbox attack "Prior convictions: Black-box adversarial attacks with bandits and priors" by leveraging gradients from substitute models. In details, they first train several reference models. When generating adversarial examples, gradients from one reference model will be randomly picked to calculate the search directions. Gradients w.r.t the input image will be approximated based on the calculated search directions. Finally, the adversarial examples are generated through iterative-FGSM/PGD. The proposed method successfully improved the query-efficiency and failure rate of the Bandits-TD method. The improvement comes from the gradients of reference models. However, when training reference model on ImageNet, 75,000 images from an auxiliary dataset and all the images from the ImageNet validation set are used. One concern is that in reality, the adversary may not have access to such large amount of auxiliary images and validation images. Without those images, it will be hard to train a useful reference model thus cannot improve the query-efficiency of blackbox attack. It seems that such efficiency does not come from the method but from extra information. Originality: the idea is not brand new. Works with similar ideas are available online, such as "Improving Black-box Adversarial Attacks with a Transfer-based Prior". Quality: The method is technically sound and the evaluation is fair. But it is not supported by theoretical analysis. Clarity: The paper is clearly written and well-organized. Adequate information and code are provided for reproduction. Significance: Improving the query-efficiency of score-based attack is good but it depends on the availability of extra information.

[Author Response · NeurIPS 2019]

We would like to thank all the reviewers for recognizing the contributions of our work and providing valuable feedback. Below are our responses to the comments.

# 1 To Reviewer #1

To the comment ". . . some more direct evidence to support the key arguments. . . ": we follow the kind suggestion from the reviewer and compare the cosine similarity between prior and true gradients in absolute values with that evaluated using Gaussian vectors. We tested on 100 CIFAR-10 images with a WRN as the victim model, and the average results are: 0.152 for ResNet-56, 0.154 for ResNet-32, 0.130 for VGG-19, 0.141 for VGG-16, and 0.014 for random Gaussian vectors. Apparently, prior gradients calculated on these reference models are an order of magnitude more similar to the true gradient of WRN, than random Gaussian vectors.

To the comment ". . . Figure 1 and subspace attack. . . ": we would like to explain that our idea stems from performing zeroth-order optimizations in linear subspaces with reduced dimensionalities, as introduced in the first half of Section 3 in our paper. Based on prior evidence, we hypothesize that different DNN models may share similar input-gradient (subspaces) and it can be more principled to substitute the random basis vectors with prior gradients (line 116-119). At first, these basis vectors are set to be *fixed* over the entire optimization procedure, that said, only the input-gradient of reference models with respect to the clean image $x$ is utilized. Apparently, then a natural extension/improvement is to *adaptively* calculate the prior gradients (of the reference models) with respect to the current

Figure 1: Choices of basis vectors.

estimation of adversarial example $x_{adv}$, i.e., our solution introduced in the second half of Section 3. See Figure 1 for an illustrative comparison of different choices of basis vectors. Hope this also addresses the minor concern regarding "adaptive". We will revise the paper accordingly and add more explanations to enhance the clarity and readability.

To the comment ". . . only if we know the data distribution first": we now apply our method under more severe domain shifts to address your concern. We attempt to train reference models in significantly different domains from that of the target. Specifically, we use reference models trained on 1) ImageNet and 2) noisy CIFAR-10.1 images (with additive Gaussian noise and $\sigma = 10/255$) respectively to attack a victim WRN on CIFAR-10, and still obtain ∼33% and ∼30% reductions in query on the base of Bandits-TD, both with lower failure rates.

To the comment ". . . why it is coordinate descent": we use coordinate descent there to indicate optimization procedures that search along the direction of one basis vector at each iteration, in contrast to the procedures whose update directions do not necessarily align with single basis vectors. Note that, unlike in a Cartesian coordinate system whose basis vectors are orthogonal one-hot vectors, ours are some prior gradients thus it is slightly different from the scenario of ZOO.

# 2 To Reviewer #2

We appreciate the suggestion about introducing a new threat model, and we agree that it is realistic to consider the cost of training reference models when performing attacks. We will carefully explain and comment on the suggested threat model in an updated version of this paper for comparison fairness. Yet, here we would also like to mention softly that it might be a bit subjective to evaluate the cost of the reference models, since querying a victim model can be sensitive [1] and costly (in money), depending on where it is hosted. Specifically, when targeting at a system which is expensive to query, an adversary may still tend to train reference models even if the number of target points is small.

# 3 To Reviewer #3

To the comment ". . . may not have access to such large amount of auxiliary images. . . ": we explain that our method does not always require a large number of images to train reference models. It is shown in Table 2 in our paper how the number of auxiliary images would affect the attack failure rate and query efficiency. While more auxiliary images are always preferred to enhance the reference models and further reduce the query complexity for our method, thousands (or even hundreds of) images can still be beneficial to achieve superior performance to previous state-of-the-arts.

To the comment ". . . similar ideas are available online. . . ": we appreciate the pointer to this contemporaneous work. It seems that the work is available online after the conference submission deadline, and we shall discuss about it in an updated version. Our experimental results are more significant than theirs in two aspects: 1) we utilize far less training images (only 75K) for ImageNet that are strictly unseen by the victim model to obtain our reference models while the standard training set (with 1.2M images) is adopted in the contemporaneous work, which means our training cost is also much cheaper, 2) we demonstrate that our method outperforms existing competitors in both targeted and untargeted settings while it only considers the untargeted setting, (note that as pointed in some papers [2], the ineffectiveness under a targeted setting can be a main drawback in transfered-based attacks).

# References

[1] S. Chen, N. Carlini, and D. Wagner. Stateful detection of black-box adversarial attacks. *arXiv preprint arXiv:1907.05587*, 2019.

[2] Y. Liu, X. Chen, C. Liu, and D. Song. Delving into transferable adversarial examples and black-box attacks. In *ICLR*, 2017.


[Meta-Review · NeurIPS 2019]

The paper proposes to use substitute models to improve the query count of black box attacks. All the reviewers agreed that this is an interesting paper although the proposed approach lacks theoretical justification. We encourage the authors to properly discuss the potential cost of training reference models and discuss whether that will be too expensive for large data (e.g., ImageNet) in the final version.